# Determination of In Vitro Antimicrobial Susceptibility for Lefamulin (Pleuromutilin) for *Ureaplasma* Spp. and *Mycoplasma hominis*

**DOI:** 10.3390/antibiotics10111370

**Published:** 2021-11-09

**Authors:** Oliver Spiller-Boulter, Susanne Paukner, Ian Boostrom, Kirsty Sands, Edward A. R. Portal, Owen B. Spiller

**Affiliations:** 1HealthFirst Consulting, Research Division, Blackwood NP12 3RF, UK; oliverspillerboulter@gmail.com; 2Nabriva Theraputics AG, 1110 Vienna, Austria; susanne.paukner@nabriva.com; 3Division of Infection and Immunity, Department of Medical Microbiology, Cardiff University, 6th Floor University Hospital of Wales, Cardiff CF14 4XN, UK; Boostrom@cardiff.ac.uk (I.B.); kirsty.sands@zoo.ox.ac.uk (K.S.); PortalE@cardiff.ac.uk (E.A.R.P.); 4Department of Zoology, University of Oxford, Oxford OX1 3RE, UK; 5Bacterial Reference Department, UK Health Security Agency, London NW9 5DF, UK

**Keywords:** lefamulin, *Mycoplasma hominis*, *Ureaplasma* spp., pleuromutilin, susceptibility testing

## Abstract

Lefamulin is the first of the pleuromutilin class of antimicrobials to be available for therapeutic use in humans. Minimum inhibitory concentrations of lefamulin were determined by microbroth dilution for 90 characterised clinical isolates (25 *Ureaplasma* *parvum*, 25 *Ureaplasma urealyticum,* and 40 *Mycoplasma hominis*). All *Mycoplasma hominis* isolates possessed lefamulin MICs of ≤0.25 mg/L after 48 h (MIC_50/90_ of 0.06/0.12 mg/L), despite an inherent resistance to macrolides; while *Ureaplasma* isolates had MICs of ≤2 mg/L after 24 h (MIC_50/90_ of 0.25/1 mg/L), despite inherent resistance to clindamycin. Two *U. urealyticum* isolates with additional A2058G mutations of 23S rRNA, and one *U. parvum* isolate with a R66Q67 deletion (all of which had a combined resistance to macrolides and clindamycin) only showed a 2-fold increase in lefamulin MIC (1–2 mg/L) relative to macrolide-susceptible strains. Lefamulin could be an effective alternative antimicrobial for treating *Ureaplasma* spp. and *Mycoplasma hominis* infections irrespective of intrinsic or acquired resistance to macrolides, lincosamides, and ketolides. Based on this potent in vitro activity and the known good, rapid, and homogenous tissue penetration of female and male urogenital tissues and glands, further exploration of clinical efficacy of lefamulin for the treatment of *Mycoplasma* and *Ureaplasma* urogenital infections is warranted.

## 1. Introduction

The class *Mollicutes* is composed of bacteria that do not make the peptidoglycan wall used to characterise bacteria by Gram staining into “Gram-positive” or “Gram-negative” isolates and therefore default to a “Gram-neutral” designation [1]. The lack of this structural feature, caused by reductive genome evolution [2], has also resulted in inherent resistance to the entire beta-lactam/carbapenem class of antimicrobials [3]. The family, *Mycoplasmataceae,* contains human and animal pathogens of the respiratory and genital tract that encompass the genera, *Mycoplasma* and *Ureaplasma.* The generalised term, human urogenital mycoplasma infections, encompass four species (*M. genitalium*, *M. hominis*, *U. parvum* and *U. urealyticum*) that are associated with symptomatic genital infections and infertility but can also be present as asymptomatic infections [4,5,6,7].

Effective therapeutics for treating symptomatic infections are further reduced for genital mycoplasmas by the lack of nucleotide synthesis (nucleotide scavenging means no effect of trimethoprim/sulfamethoxazone [8]) and facultative anaerobe capacity of these bacteria (at least for human genital mycoplasmas, which are not susceptible to gentamicin [9], but also not sensitive to the anaerobic bacterial therapeutic metronidazole [10]). Pleuromutilins have been exclusively used to treat veterinary infections to block bacterial protein synthesis by binding the peptidyl transferase centre [11]. Antimicrobial resistance mechanisms identified for therapeutics that overlap this binding site (e.g., oxazolidinones, lincosamides, phenicols, and streptogramins) may mediate resistance (or cross-resistance) to pleuromutilins. These have been reviewed by Paukner and Reidl in [12] and include ABC-F transporter or methylating resistance genes found on mobile elements (plasmids and integral conjugative elements) or mutations in the 23S rRNA or L3 and L4 ribosomal proteins. The pleuromutilin family has been shown to be effective against a wide range of relevant veterinary mycoplasma infections (reviewed by Pereyre and Tardy in this Special Issue [13], but lefamulin has recently been licenced for use in treating community-acquired bacterial pneumonia in adults. Large surveillance studies have investigated predominantly the lefamulin susceptibility of organisms causing community-acquired respiratory tract infections [14,15,16]. Studies also included antibiotic-resistant isolates, such as macrolide-resistant *Mycoplasma pneumoniae* isolates and methicillin-resistant *S. aureus* [17,18]. More recently, the in vitro susceptibility of *M. genitalium*, *Neisseria gonorrhoeae*, and other sexually transmitted bacterial pathogens were evaluated, demonstrating that lefamulin may be a promising first-line antimicrobial for the treatment of multi-drug resistant STIs [19]. However, it has not been evaluated against *Ureaplasma* spp, which are inherently resistant to all lincosamides, nor *M. hominis*, which are inherently resistant to all macrolides, resulting from conserved 23S rRNA polymorphisms [20].

We wished to determine if these differential susceptibilities to MLSK antimicrobials extended to differences in susceptibility to lefamulin. Here, we determine the minimum inhibitory concentration (MIC) of lefamulin against a panel of characterised clinical genital mycoplasma isolates including fully-genome sequenced multi-drug resistant strains resistant to macrolides, lincosamides, and ketolides. Resistance to lincosamides was linked to reduced lefamulin susceptibility, but macrolide resistance (alone or in combination with lincosamide resistance) had little effect on lefamulin MIC values. Furthermore, all isolates examined would have likely been successfully treated by lefamulin, as the MIC values were lower than the trough levels observed for lefamulin pharmacokinetics in human patients.

## 2. Results

Lefamulin and comparators were tested against 40 different clinical isolates of *M. hominis*. All *M. hominis* strains were inherently resistant to erythromycin and azithromycin (data not shown), mediated by the previously published inherent conserved polymorphism G2057A transition in their 23S rRNA sequences [20]. This cohort also included multi-drug resistant strains (Urogen006 and MH15-3) that are additionally resistant to all fluoroquinolones (mediated by S83L *gyrA* and S81I *parC* mutations) and tetracyclines (mediated by *tet*M) [21]. Susceptibility testing against lefamulin found no isolate to have an MIC >0.25 mg/L after 48 h incubation (range of 0.03–0.25 mg/L; MIC_50_ of 0.06 and MIC_90_ of 0.12 mg/L) (Table 1). The activity of lefamulin was comparable to moxifloxacin and clindamycin, with no impact of quinolone resistance (mediated by gyrase and topoisomerase mutations) on lefamulin susceptibility; however, no clindamycin resistant isolates were available to investigate.

Lefamulin and comparators were also tested against 50 unique strains of *Ureaplasma* spp. (25 *U. parvum* and 25 *U. urealyticum*). These strains included 12 of the 14 prototype strains deposited at ATCC, while the remainder were fully characterised clinical strains that included a range of tetracycline (mediated by *tet*M), fluoroquinolone (mediated by *parC* mutations), or macrolide resistant (mediated by 23S rRNA, L4 or L22 mutations) isolates (as detailed in [22,23,24]). All *Ureaplasma* spp. are inherently resistant to lincosamides, such as clindamycin, although the underlying 23S rRNA polymorphism has not been as well defined as it has for *M. hominis*. MIC determination for lefamulin found the average MIC of *Ureaplasma* spp. (0.46 ± 0.06 mg/L) higher than for *M. hominis* (0.09 ± 0.009 mg/L; *p* < 0.01), with a range of susceptibility spanning from 0.125 to 2 mg/L (Table 1). However, the only isolates found with an MIC of 2 mg/L were two *U. urealyticum* strains that had an erythromycin MIC of 128 mg/L (strains Urogen342 and 284), both with dual 23S rRNA A2058G mutations. Four isolates had an MIC of 1 mg/L, of which three were *U. urealyticum* isolates with erythromycin MIC of 2 mg/L, but also included a *U. parvum* isolate with an erythromycin MIC of 64 mg/L and an R66Q67 deletion in the L4 ribosome accessory protein (strain O10). Two other characterised isolates with known mutations yielding an erythromycin MIC of 32 mg/L (single operon mutation of A2058G; strain ScotKnee) and one with an extra N inserted at amino acid 90 of L22 ribosome accessory protein (erythromycin MIC of 8 mg/L; strain U144), both had lefamulin MICs of 0.5 mg/L and 0.25 mg/L, respectively (Table 2). It is interesting to note that the presence of high-level macrolide resistance had a very minor effect on lefamulin susceptibility. Examining *U. parvum* isolates separately from the *U. urealyticum* strains (which was negligibly altered with or without inclusion of the macrolide-resistant isolates in analysis), found that lefamulin in vitro activity was significantly more potent for *U. parvum* (0.27 ± 0.02 mg/L) compared to *U. urealyticum* (0.49 ± 0.05 mg/L; *p* < 0.001), although the susceptibility ranges clearly overlap (Figure 1A). However, this trend was also observed for susceptibility to erythromycin (excluding resistant isolates: 0.79 ± 0.05 vs. 1.45 ± 0.15 mg/L; *p* < 0.01 Figure 1B) and tetracycline (excluding *tet*(M)-positive isolates: 0.22 ± 0.04 vs. 0.52 ± 0.08 mg/L; *p* < 0.001 Figure 1C), but not for the fluoroquinolones; levofloxacin or moxifloxacin (Figure 1D,E).

## 3. Discussion

Pleuromutilins, such as tiamulin and valnemulin, show excellent activity in veterinarian medicine against mycoplasma pathogens such as *M. hyopneumoniae*, *M. hyorhinis*, *M. hyosynoviae* (pigs) [25], *M. bovis* (cattle) [26], as well as *M. gallisepticum* and *M. synoviae* (poultry) [27]. More recently, lefamulin (Xenleta™) has been approved for the treatment of community-acquired bacterial pneumonia in the US, Europe, and other countries [28,29]. Good in vitro activity has been shown for lefamulin against macrolide-resistant variants of the respiratory mycoplasma pathogen, *M. pneumoniae* [17,18] and genital mycoplasma, *M. genitalium* [19]. This latter pathogen is of increasing concern as doxycycline is only effective in treating about 30% of infections, and prevalence of clinical strains with combined high azithromycin and moxifloxacin resistance are increasing [30]. However, neither of these human mycoplasmas are inherently resistant to macrolides or lincosamides. As pleuromutilins are known to inhibit bacterial growth by binding to the 23S rRNA, it was of interest to evaluate lefamulin susceptibilities for mycoplasmas that are inherently resistant to macrolides (*Ureaplasma* spp.) and lincosamides (*M. hominis*), as they have a limited repertoire of therapeutics and prevalence of macrolide resistance is increasing in ureaplasmas as well [31]. However, we recognised that as a limitation to our study, we were only able to assess G2057A and A2058G 23S rRNA mutations, as well as an L4 deletion and an L22 insertion mutation, on their impact for lefamulin sensitivity. 

Despite the bacterial ribosome being a major antibiotic target, only a few sites on the ribosome are known to be bound by clinically used antibiotics. On the 50S ribosomal subunit, the peptidyl transferase centre is targeted by pleuromutilins, lincosamides, streptogramin A, phenicols, and oxazolidinones, while macrolides and streptogramin B bind at adjacent sites at the beginning of the nascent peptide exit tunnel. While pleuromutilin resistance has been shown to be mediated in other genera of bacteria by plasmid-encoded or transposon-associated chromosomal resistance determinants, such as VmlR, LsaA, and VgaA [11,12], these genes have not been identified in mycoplasma genomes. However, mutations in the 23S rRNA have been identified as mechanisms of resistance. The common region of mutations is found between *E. coli* numbering 2057 to 2062, that are conserved in most mycoplasmas in the peptidyltransferase loop (domain V) of 23S (GAAAGA). The G2057A in *M. hominis* is responsible for the inherent macrolide resistance, while A2058G in *Ureaplasma* spp. is responsible for acquired macrolide resistance. Acquired macrolide resistance in *M. genitalium* has been associated largely with A2058G or A2059G mutations [32], however A2058T and A2058C have been reported as well [33,34]. Resistance to veterinary pleuromutilins have been associated with mutations at A2058, A2059, A2060, G2061, and A2062, as well as mutations at G2447, U2500, and A2503 (Pereyre and Tardy reviewed in this Special Issue [13]); however, A2058G mutation for *U. urealyticum* in our study was found to not affect the MIC of lefamulin. In *M. gallisepticum* studies of resistance to tiamulin and valnemulin, it was found that A2503U was most common, resulting in a 64-fold increase in tiamulin MIC, and this was further increased to ≥250-fold increase if combined with a G2061U or G2447A mutation [35]. To date, no resistance to lefamulin has been found for any human mycoplasma tested, including macrolide resistant *M. pneuomoniae* and *M. genitalium* strains examined in other reports [18,19]; however, it is unknown if the mutations G2061U, A2503U, or G2447A occur in human mycoplasmas.

Lefamulin susceptibility breakpoints have not yet been established for *Mycoplasma* spp. and *Ureaplasma* spp. by the relevant international bodies. Noteworthy, the clinical efficacy of oral or intravenous lefamulin in patients with community-acquired bacterial pneumonia and *M. pneumoniae* at baseline in phase 3 clinical trials, was high (early clinical response of 92.3–96.6%) in various analysis populations [14,15,16]. Measurement of lefamulin exposure in plasma and target attainment analyses using epithelial lining fluid (ELF) and free drug plasma concentrations, clinical efficacy, and pharmacokinetic/pharmacodynamic (PK/PD) analyses, led to susceptible breakpoints of ≤0.5 mg/L for *S. pneumoniae*, ≤0.25 mg/L for *S. aureus*, and ≤2 mg/L for *H. influenzae* [17], concentrations that might also cover a high proportion of *M. hominis* and *Ureaplasma* spp. isolates. In addition, evaluation of lefamulin exposure in the urogenital tract in rats demonstrated rapid and extensive penetration into female and male tissues and glands, similar to or greater than that observed in the lung [36].

In summary, we found *M. hominis* to be particularly susceptible to lefamulin, with a maximum MIC of 0.25 mg/L, regardless of resistance to 14- and 15-membered macrolides inherent to this species. *Ureaplasma* spp., which are inherently resistant to lincosamides, were still highly susceptible to lefamulin but less susceptible than *M. hominis*. Further, *Ureaplasma urealyticum* strains that had A2058G 23S rRNA mutations (mediating additional high erythromycin resistance) showed only a 2-fold increase in lefamulin MIC (maximum 2 mg/L), which is still below the trough levels shown for pharmacokinetic studies in humans.

## 4. Conclusions

Lefamulin may be an effective alternative antimicrobial for treating *Ureaplasma* spp. and *M. hominis* infections, since it demonstrated potent in vitro antibacterial activity against *Ureaplasma* spp. and *Mycoplasma hominis*, irrespective of intrinsic or acquired resistance to macrolides, lincosamides, and ketolides. 23S rRNA or L4 mutations known to give high macrolide resistance, resulted in a 2-fold increase in the MICs of lefamulin relative to the highest MICs for macrolide sensitive strains of the same species, while *M. hominis,* that is intrinsically resistant to macrolides, was highly susceptible to lefamulin. Further evaluation of the efficacy of lefamulin for the treatment of *Mycoplasma* and *Ureaplasma* urogenital infections is warranted.

## 5. Materials and Methods

### 5.1. Reference Isolates

A panel of 90 isolated strains (25 *Ureaplasma parvum*, 25 *Ureaplasma urealyticum*, and 40 *M. hominis*), characterised from previously published clinical cohorts [21,22,23,24] as well as from the UROGEN study (IRAS 251053; Spiller O.B. and Jones L.C. unpublished), with known antimicrobial susceptibility or characterised resistance, were used. These isolates were initially typed by multiplex quantitative PCR that targets conserved sequences in the *yidC* gene for *M. hominis* and species-specific variations in the *ureC* gene for *U. parvum* and *U. urealyticum,* as previously published [24]. Included in the cohort were 12 tetracycline-resistant *M. hominis*, 2 combined fluoroquinolone- and tetracycline-resistant *M. hominis*, 1 moxifloxacin-resistant *U. parvum*, 5 levofloxacin-resistant *U. parvum*, 1 macrolide-resistant *U. parvum,* and 3 dual macrolide- and ketolide-resistant *U. urealyticum* isolates.

### 5.2. Antimicrobial Susceptibility Testing (AST) Method

Minimum inhibitory concentrations (MICs) were determined for all isolates using broth microdilution methods, compliant with CLSI M43A guidelines [37,38]. Experiments were conducted in duplicate (technical replicate) and confirmed for the same isolates on a separate day (biological replicate) to ensure integrity of the findings. Resistance thresholds (or breakpoints) are internationally set as follows: for *Ureaplasma* spp. moxifloxacin and levofloxacin ≥4 mg/L, tetracycline ≥2 mg/L, and erythromycin ≥16 mg/L; for *M. hominis* levofloxacin ≥2 mg/L, moxifloxacin ≥4 mg/L, tetracycline ≥8 mg/L, and clindamycin ≥0.5 mg/L. No resistance thresholds or breakpoints have been set for lefamulin; therefore, the susceptibility was tested for all isolates over a range of 0.0156 to 64 mg/L. *Ureaplasma* spp. was grown in Ureaplasma Selective Medium, purchased from Mycoplasma Experience Ltd. (Reigate, UK) and *M. hominis* was grown in Lyo2 medium, purchased from bioMérieux (France). Antibiotics were purchased from Sigma Aldrich (Poole, UK) except for lefamulin, which was provided by Nabriva Therapeutics GmbH (Vienna, Austria). MICs were performed as previously published [23] using the broth microdilution method in 96-well plates, sealed under normoxic conditions, and incubated at 37 °C. Plates were examined at regular intervals to ensure viable growth of strains in the antimicrobial-free growth control wells, and MIC results were recorded for *Ureaplasma* spp. at 24 h and for *M. hominis,* at 48 h post-incubation. Bacterial growth was visualised by a colour change from yellow to red of the medium (mediated by phenol red pH indicator) and MICs read for the rows containing 10,000 colour changing units in the antimicrobial-free growth control row at the bottom of the plate.

### 5.3. Statistical Analysis

Significance was set at *p* < 0.05 and analysis was carried out for each species independently using GraphPad Prism (version 8) to determine basic descriptive statistics (mean, standard deviation of the mean, CI_95_, etc.). Comparison of MIC ranges between *U. parvum* and *U. urealyticum* by unpaired t-test, or *M. hominis* compared to *Ureaplasma* spp., where MIC for all three species was compared, a one-way ANOVA with Tukey’s post hoc testing (with correction for multiple comparisons) was used.

### 5.4. Whole Genome Sequencing

Whole genome sequencing had previously been performed for 26 *M. hominis* strains (including those submitted under BioProject number PRJNA675754), 25 *U. parvum* strains (including one submitted under BioProject number PRJNA767530, as well as ATCC prototype serovars 1, 3, 6, and 14, respectively, submitted under Bioprojects (PRJNA20245, PRJNA19087, PRJNA19413 and PRJNA20185), and 21 *U. urealyticum* strains (including those submitted under BioProject number PRJNA767530), as well as ATCC prototype serovars 2, 4, 5, 7, 8, 9, 11, and 12, respectively, submitted under BioProjects (PRJNA20687, PRJNA19091, PRJNA19415, PRJNA19093, PRJNA19089, PRJNA19095, PRJNA19417, and PRJNA19419). Short read sequencing was performed on an Illumina MiSeq with scaled up cultures of *Ureaplasma* and *Mycoplasma* strains, following genomic DNA extraction, as described in previously published methods [21,24]. Shovill (v0.9.0) was used as a de novo assembler (no reference genome), and sequence quality was determined using QUAST (v5.0.2), except for those WGS for ATCC deposited strains and for macrolide resistant *Ureaplasma* spp. Long-read sequencing on a MinION device (Oxford Nanopore Technology) was additionally performed for the macrolide resistant *Ureaplasma* spp. strains O10, ScotKnee, U144, and Urogen342, using the same gDNA extracts for short read sequencing. Sealed whole genomes for each isolate were generated by combined analysis of long and short read sequences using Unicycler (v0.4.7) and the resultant genomes have been submitted as BioProject PRJNA767530.

## Figures and Tables

**Figure 1 antibiotics-10-01370-f001:**
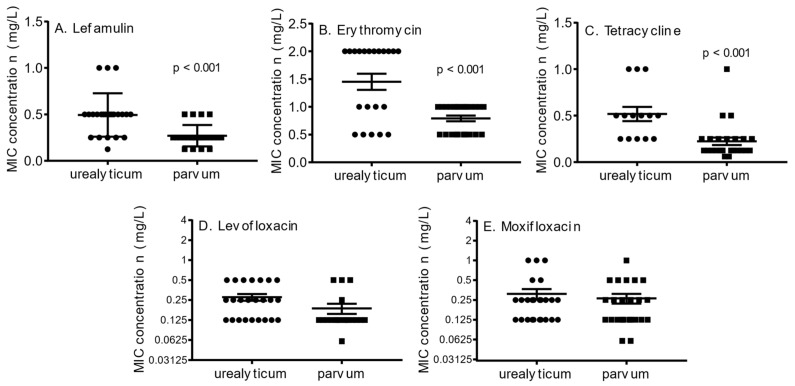
Minimal inhibitory concentrations for *U. urealyticum* relative to *U. parvum* for (**A**) lefamulin, (**B**) erythromycin, (**C**) tetracycline, (**D**) levofloxacin, and (**E**) moxifloxainafter strains with 23S rRNA A2058G, L4 and L22 mutations have been removed (**A**,**B**), strains carrying the gene *tet*(M) have been removed (**C**), or strains carrying mutations in the quinolone resistance determining region of *parC* and *gyrB* have been removed (**D**,**E**). Mean and standard error of the mean are also shown, with unpaired *t*-test analysis.

**Table 1 antibiotics-10-01370-t001:** MIC (mg/L) distributions of lefamulin and comparators for *Ureaplasma* spp. (n = 50) and *Mycoplasma hominis* (n = 40).

Organism and Antimicrobial	0.015	0.03	0.06	0.12	0.25	0.5	1	2	4	8	16	32	64	128	MIC_50_	MIC_90_
*Ureaplasma* spp. *n* (cumulative %)																
**Lefamulin**				5	21	18	4	2							0.25	1
				(10)	(52)	(88)	(96)	(100)								
**Levofloxacin**				23	9	10	1	3	**1**			**3**			0.25	2
				(46)	(64)	(82)	(86)	(92)	**(94)**			**(100)**				
**Moxifloxacin**			2	12	11	5	4			**4**					0.25	
			(5.3)	(36.8)	(65.8)	(78.9)	(89.5)			**(100)**						
**Tetracycline**			2	13	24	8	4				**2**	**2**	**10**		0.5	64
			(4)	(26)	(48)	(64)	(72)				(76)	(80)	(100)			
**Erythromycin**						15	18	12		1		**1**	**1**	**2**	1	2
						(30)	(54)	(90)		(92)		(94)	(96)	(100)		
**Azithromycin**				4	8	2		1		1		2			1	
				(22.2)	(66.6)	(77.7)		(83.3)		(88.8)		(100)				
**Telithromycin**		2	3	4	1		1				2				0.12	
		(15.4)	(38.7)	(69.2)	(76.9)		(84.6)				(100)					
*M. hominis, n* (cumulative %)																
**Lefamulin**		8	16	14	**2**										0.06	0.12
		(20)	(60)	(95)	(100)											
**Levofloxacin**					16	11	1			**2**					0.5	0.5
					(39)	(92.5)	(95)			100						
**Moxifloxacin**		2	32	2	1	1			**2**						0.06	0.12
		(5)	(85)	(90)	(92.5)	(95)			(100)							
**Tetracycline**				20		13	4				**4**	**7**	**2**		0.5	32 ^a^
				(50)		(57.5)	(67.5)				(77.5)	(95)	(100)			
**Clindamycin**	**2**	14	12	6											0.06	0.12
	(5)	(39)	(85)	(100)												

The number of isolates is listed from lowest to highest dilution, with the cumulative percentage shown in brackets below. The isolates at or above the resistance breakpoints are shown in bold. MICs of antimicrobials were determined using broth microdilution methods as defined by the CLSI publication M43. ^a^ The MIC_90_ for tetracycline is skewed due to the large number of *tet*(M)-containing reference and endogenous strains present. Excluding *tet*(M) strains, the MIC_90_ would be 0.5 mg/L.

**Table 2 antibiotics-10-01370-t002:** Lefamulin susceptibility of *Ureaplasma* spp. with elevated erythromycin MIC.

Strain	Species	Erythromycin MIC (mg/L)	Lefamulin MIC (mg/L)	23S rRNA Copy 1 ^a^	23S rRNA Copy 2 ^b^	L4	L22
Urogen284	*U. urealyticum*	128	2	A2058G	A2058G	N ^c^	N
Urogen324	*U. urealyticum*	128	2	A2058G	A2058G	N	N
ScotKnee	*U. urealyticum*	32	0.5	A2058G	N	N	N
O10	*U. parvum*	64	1	N	N	Del R66Q67	N
U144	*U. urealyticum*	8	0.25	N	N	N	Ins N90

There are two copies of 23S rRNA in all *Ureaplasma* spp. genomes: ^a^ 23S rRNA copy adjacent to the pseudouridine synthase gene and ^b^ 23S rRNA copy adjacent to the transcription elongation factor GreA. ^c^ No resistance-mediating mutation found.

## Data Availability

Data available from the authors upon reasonable request.

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
