# Peer review of "Determination of In Vitro Antimicrobial Susceptibility for Lefamulin (Pleuromutilin) for Ureaplasma Spp. and Mycoplasma hominis"

_antibiotics, 2021, doi:10.3390/antibiotics10111370_

Round 1
Reviewer 1 Report
The MS by Oliver SPILLER-BOULTER et al talks about in vitro evaluation of antimicrobial susceptibility for Lefamulin (pleuromutilin) for Ureaplasma spp. and Mycoplasma hominis. The resistance on Mollicutes to various conventional antimicrobials lead to severe problems in infections treatment and the proposed solution, i.e. Lefamulin to target these pathogens looks as promising hope for the future.
While the ms is well written, easy to read and experiments are well planned, some suggestions should be adressed to authors.
Intro:
please
- add more references (best - reviews) to general characteristics of these bacteria
- authors should describe broader native mechanisms of resistance to antimicrobials
- please add the aim, short description of work and conclusions at the end of intro.
Results paragraph 2. Authors mention various mutations leading to resistance. Please refer to data where it has been shown (papers, seq data)
In the end of discussion, please highlight the main findings
i suggest to move either conclusions before methods, or methods before results (better)
Methods:
please describe how isolates were typed
Author Response
Intro:
please
- add more references (best - reviews) to general characteristics of these bacteria
We have now added more references (mostly reviews) to substantiate the general characteristics of these bacteria in the introduction as requested.
- authors should describe broader native mechanisms of resistance to antimicrobials
We have added details of potential resistance mechanisms to pleuromutilins and cited a comprehensive review on the topic.
- please add the aim, short description of work and conclusions at the end of intro.
This has now been added.
Results paragraph 2. Authors mention various mutations leading to resistance. Please refer to data where it has been shown (papers, seq data)
We have now added more information and the appropriate citations for where these isolates have been previously characterised.
In the end of discussion, please highlight the main findings
We have added a summarising paragraph highlighting the key findings.
i suggest to move either conclusions before methods, or methods before results (better)
Style for the journal requires methods to be last, so we have moved the conclusions before the methods.
Methods:
please describe how isolates were typed
The isolates were typed by previously clinically validated qPCR, which has now been added and appropriately cited in the methods.
Reviewer 2 Report
Many conventional antibiotics used in clinical medicine have failed to treat bacterial infections. Because of this, studies such as the one described by Spiller-Boulter and collaborators are relevant and open new paths for advances in antibacterial therapy. Lefamulin showed an in vitro effect against Ureaplasma spp. and Mycoplasma hominis. Minor suggestions should be reviewed by the authors.
- Some details are important in the methodology section. Was the experiment (MIC determination) repeated? How many times?
- According to the authors, microbial growth was monitored by color change. How did the authors register this change? Did they use spectrophotometry?
Author Response
- Some details are important in the methodology section. Was the experiment (MIC determination) repeated? How many times?
We performed the MIC determinations in duplicate (technical replicates) and also confirmed the findings for the strains on a separate day (biological replicate). We have clarified this in the materials and methods.
- According to the authors, microbial growth was monitored by color change. How did the authors register this change? Did they use spectrophotometry?
The colour change for media is related to pH as detected by phenol red, as such we are assessing a bright yellow colour to a distinctively different red colour, which does not require a spectrophotometer to determine. This has been clarified in the materials and methods.
Reviewer 3 Report
Review of the Manuscript ID: antibiotics-1424072 entitled Determination of in vitro antimicrobial susceptibility for Lefamulin (pleuromutilin) for Ureaplasma spp. and Mycoplasma hominis
Good research, properly prepared Manuscript. However, I have some observations:
General:
➢ Important and necessary research, the methodology does not raise any objections, interesting results, but I do not know if the number of analyzes and results is sufficiently wide for a good publication...
➢ Please check the punctuation throughout the Manuscript – spaces, parentheses... Many such minor errors throughout the Manuscript...
➢ Please standardize the units – once the Authors give the full name and once an abbreviation (e.g. hour and hr)
➢ Taking into account the small number of results, maybe it is worth including the results, which the authors refer to in the text as “data not shown”?

Author Response
General:
➢ Important and necessary research, the methodology does not raise any objections, interesting results, but I do not know if the number of analyzes and results is sufficiently wide for a good publication...
We thank the reviewer for their observations. We performed the analyses in duplicate (technical replicates) and also confirmed the findings for strains on a separate day (biological replicate). We have clarified this in the materials and methods. With respect to sample size, previous publication for in vitro susceptibility on Lefamulin that was published in the highly regarded Antimicrobial Agents and Chemotherapy on other STI bacteria included on average fewer isolates (Chlamydia trachomatis (n = 15), susceptible and multidrug-resistant Mycoplasma genitalium (n = 6), and susceptible and resistant Neisseria gonorrhoeae (n = 25). While our study was comparable to the number of isolates, also published in AAC, for in vitro susceptibility testing for Mycoplasma pneumoniae (18 macrolide-susceptible and 42 macrolide-resistant Mycoplasma pneumoniae strains).
➢ Please check the punctuation throughout the Manuscript – spaces, parentheses... Many such minor errors throughout the Manuscript...
We have re-read the manuscript to ensure minor errors have been eliminated.
➢ Please standardize the units – once the Authors give the full name and once an abbreviation (e.g. hour and hr)
We apologise for in consistent use and have gone through to ensure consistency for abbreviation use.
➢ Taking into account the small number of results, maybe it is worth including the results, which the authors refer to in the text as “data not shown”?
We have included the moxifloxacin and levofloxacin data as requested, however we haven’t included the data confirming all M. hominis strains are resistant to erythromycin and azithromycin as the values are >128 and .>16, which cannot be made into an interesting figure or table.
Reviewer 4 Report
p.2 , L1 “ …… and facultative anaerobe capacity of these bacteria (no susceptibility to aminoglycosides…” – this statement is a bit problematic since several veterinary Mycoplasma spp are susceptible, in general, to aminoglycosides (there is no intrinsic resistance to this class of antibiotics).
p. 6, L46 “…however A2058G mutation for U. urealyticum in our study was found to not affect the MIC of lefamulin MIC.” it is worth to add that only limited number of macrolide resistant strains /strains with macrolide-defined genotypes have been tested in this study .
Author Response
p.2 , L1 “ …… and facultative anaerobe capacity of these bacteria (no susceptibility to aminoglycosides…” – this statement is a bit problematic since several veterinary Mycoplasma spp are susceptible, in general, to aminoglycosides (there is no intrinsic resistance to this class of antibiotics).
We apologise for this oversight; we were unaware of this. We have corrected this statement to reflect what we know to be true for human genital mycoplasmas and their response to gentamicin.
- 6, L46 “…however A2058G mutation for U. urealyticum in our study was found to not affect the MIC of lefamulin MIC.” it is worth to add that only limited number of macrolide resistant strains /strains with macrolide-defined genotypes have been tested in this study .
We agree with the reviewer, there are more possibilities regarding pleuromutilin resistance than we we had available to us to test. We have included this as a limitation as requested.
Round 2
Reviewer 1 Report
Authors adressed al questions